# THE DEVIL IS IN THE WORD: VIDEO-CONDITIONED TEXT REPRESENTATION REFINEMENT FOR TEXT-TO-VIDEO RETRIEVAL

## ABSTRACT

Pre-trained vision-language models (VLMs), such as CLIP, have shown remarkable success in the text-video retrieval task due to their strong vision-language representations learned from large-scale paired image-text samples. However, compared to videos, text is often short and concise, making it difficult to fully capture the rich and redundant semantics present in a video with thousands of frames. Recent advances have focused on utilizing text features to extract key information from these redundant video frames. However, text representation generated without considering video information can suffer from bias and lack the expressiveness needed to capture key words that could enhance retrieval performance. In this study, we first conduct preliminary experiments to demonstrate the importance of enhancing text representations. These experiments reveal that text representation only generated from text input often misinterpret critical information. To address this, we propose a simple yet efficient method, VICTER, *i.e.*, **vi**deo-**c**onditioned **te**xt representation **r**efinement, to enrich text representation using a versatile module. Specifically, we introduce a video abstraction module that extracts representative features from multiple video frames. This is followed by a video-conditioned text enhancement module that refines the original text features by reassessing individual word features and extracting key words using the generated video features. Empirical evidence shows that VICTER not only effectively captures relevant key words from the input text but also complements various existing frameworks. Our experimental results demonstrate a significant improvement of VICTER over several baseline frameworks (with $0.4\% \sim 1.0\%$ improvements on R@1). Furthermore, VICTER achieves state-of-the-art performance on three benchmark datasets, including MSRVTT, DiDeMo, and LSMDC. Code will be made available.

## 1 INTRODUCTION

Text-video retrieval aims to find the most relevant video or text based on a given query (Gabeur et al., 2020; Luo et al., 2022; Zhu et al., 2023a; Ging et al., 2020). With the exponential growth of online video content on platforms like YouTube, Netflix, and TikTok, text-video retrieval has gained increasing attention and plays a vital role in modern applications, such as search engines and recommendation systems (Davidson et al., 2010; Gomez-Uribe & Hunt, 2015).

The emergence of vision-language models (VLMs) like CLIP (Radford et al., 2021), which are pretrained on large-scale image-text datasets, offers a powerful solution to this problem. These models have demonstrated remarkable cross-modal representation capabilities, making it feasible to transfer their knowledge from images to videos (Luo et al., 2022; Gan et al., 2023). Specifically, CLIP uses a dual-branch structure, comprising a text encoder and an image encoder, to learn cross-modal alignment. However, CLIP's image encoder can only generate frame-level features, which fail to capture the temporal information inherent in videos. As a result, one research direction has been to explore better ways to leverage this temporal information (Bain et al., 2021; Liu et al., 2022).

A video typically contains hundreds of frames, while text descriptions—such as captions or subtitles—are often much shorter and consist of only a few words (Lin et al., 2022; Wang et al., 2024).

This imbalance between the vast amount of visual data and the concise nature of textual descriptions poses a significant challenge for accurate retrieval. Consequently, another key trend in the field has focused on using text to identify key information in videos while filtering out redundant or noisy frames (Guan et al., 2023; Gorti et al., 2022; Jin et al., 2023a).

While much of the prior research has concentrated on improving video representations, we argue that refining the text representation is equally important, yet has been underexplored. Through a preliminary experiment, we demonstrate that a more concise and precise text description can significantly enhance retrieval performance. Crucially, this refinement can only be effective when paired with corresponding video information. Instead of relying on users to provide long and detailed text descriptions in real-world applications, we aim to refine the text representation itself using video context, ensuring that the enhancement happens at the feature level rather than altering the original text. Moreover, we observe that current methods, particularly those based on CLIP, exhibit a bias towards nouns in text descriptions. This bias can lead to suboptimal results, as key aspects of the text—such as verbs and even prepositions—are often overlooked, despite being crucial for retrieval. Inspired by this observation, we propose a simple yet effective solution, VICTER, *i.e.*, **vi**deo-**c**onditioned **te**xt representation **r**efinement, as shwon in Figure 1. Our approach aims to enrich text representations using video information in a flexible, modular framework.

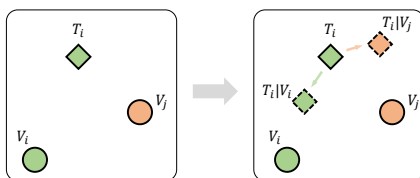

Figure 1: Illustration of the proposed VICTER, *i.e.*, **vi**deo-**c**onditioned **te**xt representation **r**efinement. Previous frameworks generate text representations without incorporating video information, which can lead to suboptimal biases inherent in pretrained VLMs. We propose leveraging video information to further refine the text representation for improving retrieval performance.

The core idea is to use the video features to re-weight the word embeddings generated by CLIP, highlighting key words that are more relevant to the video content. However, since CLIP's video features are based on frame-level representations, using simple non-parametric methods like mean pooling can dilute the relevance of the extracted features with unrelated information. To address this, we introduce a video abstraction module, which extracts a more meaningful video representation by leveraging the prior assumption that important content within a video tends to appear more frequently or occupies a larger proportion of the frames. Following this, we design a video-conditioned text enhancement module that refines the original text features by reassessing individual word embeddings and extracting the key words using the abstracted video features. During testing, this enhancement module only adds a negligible computational cost, as it operates on top of the standard text-video similarity matching process by performing an additional weighted summation over the word features. Our method is not limited to CLIP but can also be applied to other video-text frameworks (Xue et al., 2022b; Wang et al., 2023), making it a versatile approach for various retrieval tasks. By efficiently refining the text representation using video content, VICTER bridges the gap between concise text descriptions and rich video data, significantly improving retrieval accuracy with minimal additional computation. We summarize the contributions of this work as follows:

- This work demonstrates that pretrained image-text models exhibit a bias towards nouns in text descriptions. As a result, relying solely on the text encoder for feature extraction overlooks other important words, distorting the focus of the description and leading to suboptimal results. To address this issue, we propose a video-conditioned text enhancement module to leverage the video context to reassess and refine the text representation.

- We show that the primary content of a video typically occupies a larger portion of its frames. Based on this observation, we introduce a video abstraction module that integrates representative video features from frame-level data, without relying on any additional text information.

- Our VICTER is versatile and can be applied to various frameworks, including pretrained image-text and video-text models. It significantly improves performance over baselines, setting new state-of-the-art results on three benchmark datasets: MSR-VTT, LSMDC, and DiDeMo.

## 2 RELATED WORK

**Pretrained Vision-Language Model.** Vision-language pre-training aims to understand and process the relationship between image and text modalities. Early research works utilized sequence en-

coders, such as Long Short-Term Memory (LSTM) (Graves & Graves, 2012) and Gated Recurrent Units (GRU) (Chung et al., 2014), to learn language representations. With the success of BERT (Devlin, 2018) in learning contextual text representations, many vision-language works (Gabeur et al., 2020; Sun et al., 2019; Zhu & Yang, 2020; Wang et al., 2021b) began leveraging pre-trained BERT features to enhance language representation capabilities. More recently, contrastive image-text pre-training (Tan & Bansal, 2019; Radford et al., 2021) on large-scale web data has significantly advanced performance in vision-language tasks. CLIP (Radford et al., 2021), one of the most prominent pre-trained models, has demonstrated powerful zero-shot capabilities (Luo et al., 2022; Deng et al., 2023) for video-language understanding and has become a de facto baseline for text-video retrieval tasks.

**Text-to-video Retrieval.** The goal of the retrieval task is to retrieve relevant videos from a database of video clips based on a text query (Rohrbach et al., 2015; Xu et al., 2016; Wang et al., 2019; Anne Hendricks et al., 2017). Previous studies (Chen et al., 2020; Mithun et al., 2018) primarily focused on developing fusion strategies to align pre-extracted text and video features. With the introduction of CLIP, more recent approaches have concentrated on enhancing video and text representations (Croitoru et al., 2021; Lei et al., 2021; Bain et al., 2021; Liu et al., 2022; Li et al., 2023b). For example, X-Pool (Gorti et al., 2022) utilizes text-conditioned feature fusion across video frames, while PIDRo (Guan et al., 2023) models fine-grained semantic clues between video and text. UATVR (Fang et al., 2023) addresses the inherent uncertainties in both text and image modalities. DiffusionRet (Jin et al., 2023b) advances retrieval by integrating diffusion models into the text-video retrieval pipeline, and T-MASS (Wang et al., 2024) enriches text embeddings by treating them as stochastic embeddings. In contrast to previous methods focused on carefully selecting video features, our VICTER takes a complementary approach by refining the text features to address the inherent limitations of the CLIP's text encoder.

**Video-Language Post-Pretraining.** To better leverage the temporal information in video data and overcome the limitations of CLIP's image-text pretraining, several works (Wang et al., 2023; Cheng et al., 2023) re-pretrain a unified backbone architecture to directly output video-level features instead of frame-level ones. For instance, CLIP-ViP (Xue et al., 2022b) introduces a proxy-guided video attention mechanism and re-pretrains the entire framework on large datasets like WebVid-2.5M (Bain et al., 2021) and HDVILA-100M (Xue et al., 2022a) to generate richer video representations. Cap4Video (Wu et al., 2023), on the other hand, enhances video representations by incorporating auxiliary captions. Our VICTER requires no large-scale post-pretraining overhead. Instead, it serves as a plug-and-play module that can be seamlessly integrated into existing frameworks.

**Fine-grained Interaction between Words and Frames.** There is a research trend that closely relates to our VICTER, focusing on semantic alignments through fine-grained interactions (Wang et al., 2021a;b; Zhu & Yang, 2020; Ma et al., 2022) between word and frame features. However, many of these approaches introduce complex cross-modal fusion modules that emphasize specific entities within the language (*e.g.*, words and phrases) and video (*e.g.*, frames and regions) (Chen et al., 2020). While these methods have demonstrated significant performance improvements, they often come with prohibitive computational costs. In this paper, we propose a lightweight and versatile video-conditioned text enhancement module that refines the global text representation by addressing the attention bias introduced by the text encoder's emphasis on certain words.

## 3 PRELIMINARY STUDY

In this section, we first revisit the workflow for adapting pre-trained vision-language models (VLMs) to retrieval task. Next, we investigate how improving the description gap impacts performance, followed by an analysis of the challenges posed by suboptimal priors in extracting text representations.

### 3.1 PRETRAINED VLMS FOR TEXT-VIDEO RETRIEVAL

Here we introduce the generic framework for adapting pre-trained VLMs to video, and discuss how prior works fit within this framework. We use CLIP (Radford et al., 2021) as a representative VLM, given its strong performance and the availability of open-source models. CLIP comprises two encoders—one for images and one for text—that are jointly optimized on large-scale, internet-sourced image-text pairs. For a given input sentence, the text encoder produces a representation

for each word, including the $\langle \text{EOS} \rangle$ token (*i.e.*, end-of-sequence). Typically, the $\langle \text{EOS} \rangle$ token is used as the sentence embedding, denoted as $\mathbf{t}^{\langle eos \rangle} \in \mathbb{R}^d$, where $d$ represents the feature dimension. The image encoder processes each video frame, outputting frame-level representations, which we denote as $\{\mathbf{v}^1, \mathbf{v}^2, ..., \mathbf{v}^{\mathcal{T}}\}$. To derive the overall video representation, prior works apply either text-agnostic pooling (Luo et al., 2022) or text-conditioned pooling (Gorti et al., 2022), resulting in a pooled video feature $\mathbf{v}^{\text{pool}} \in \mathbb{R}^d$. For retrieval, a similarity function $s(\cdot)$, such as cosine similarity, is employed to measure the relevance between text and video features. Given a training dataset with $N$ distinct text-video pairs, $\mathcal{D} = \{(t_i^{\langle \text{eos} \rangle}, v_i^{\text{pool}})\}_{i=1}^N$, the model is optimized using the InfoNCE loss (Oord et al., 2018), where both text-to-video ($\mathcal{L}t \rightarrow v$) and video-to-text ($\mathcal{L}_{v \rightarrow t}$) retrieval losses are jointly minimized:

$$\mathcal{L}_{t \rightarrow v} = -\frac{1}{B} \sum_{i=1}^B \log \frac{e^{s(\mathbf{t}_i^{\langle \text{eos} \rangle}, \mathbf{v}_i^{\text{pool}})}}{\sum_j e^{s(\mathbf{t}_i^{\langle \text{eos} \rangle}, \mathbf{v}_j^{\text{pool}})}}, \quad \mathcal{L}_{v \rightarrow t} = -\frac{1}{B} \sum_{i=1}^B \log \frac{e^{s(\mathbf{v}_i^{\text{pool}}, \mathbf{t}_i^{\langle \text{eos} \rangle})}}{\sum_j e^{s(\mathbf{v}_i^{\text{pool}}, \mathbf{t}_j^{\langle \text{eos} \rangle})}}, \quad (1)$$

where $B$ denotes the number of text-video pairs in a batch. The overall loss combines both directions:

$$\mathcal{L}_{\text{retrieval}} = \frac{1}{2}(\mathcal{L}_{t \rightarrow v} + \mathcal{L}_{v \rightarrow t}). \quad (2)$$

This loss reaches its minimum when all relevant text-video pairs in a batch are perfectly aligned, a result that heavily relies on the quality of both the text and video representations.

## 3.2 ENHANCED TEXT FOR IMPROVED RETRIEVAL

The task of text-video retrieval involves training a model to learn a similarity function between text and video representations. However, there exists a notable discrepancy between these two modalities, as text is often short and concise, containing much less semantic information compared to its corresponding video. This makes it difficult for the text representation to fully capture the rich semantics embedded within the video.

| Method | R@1 | R@5 | R@10 |
|---|---|---|---|
| X-Pool (Gorti et al., 2022) | 46.9 | 72.8 | 82.2 |
| → Enhanced text via VLM | 57.3 (+10.4) | 79.5 (+6.7) | 88.2 (+6.0) |
| → Enhanced text via LLM | 36.2 (-10.7) | 62.4 (-10.4) | 73.0 (-9.2) |
| T-MASS (Wang et al., 2024) | 50.2 | 75.3 | 85.1 |
| → Enhanced text via VLM | 63.1 (+12.9) | 82.4 (+7.1) | 90.0 (+4.9) |
| → Enhanced text via LLM | 39.6 (-10.6) | 66.8 (-8.5) | 78.2 (-6.9) |

Table 1: Text-to-video retrieval results on MSR-VTT. Enhancing the text with relevant video significantly boosts the accuracy of existing methods.

Intuitively, we hypothesize that *enriching the text itself can significantly boost retrieval performance*. To test this hypothesis, we leverage an image captioning model, MiniGPT[1] (Zhu et al., 2023b), to generate more precise and detailed textual descriptions for the paired videos, replacing the original captions in the dataset, such as MSR-VTT (Xu et al., 2016). Some examples are shown in Figure 2, where the generated text is approximately three times longer than the original captions. We opted for an image captioning model instead of a video captioning model for three key reasons: (1) In video-text retrieval benchmarks, one video typically corresponds to multiple captions, so it's intuitive to generate diverse captions from different frames; (2) most state-of-the-art video captioning models are trained on the same video-text datasets we use (*e.g.*, MSR-VTT), so employing them could introduce data leakage, compromising fairness; and (3) the output of existing video captioning models remains far from satisfactory compared to their image captioning counterparts.

As shown in Table 1, we replaced the original text in the dataset with more informative descriptions generated using corresponding videos, while keeping the training recipe unchanged. This approach leads to substantial performance gains across different retrieval frameworks, such as X-Pool (Gorti et al., 2022) and T-MASS (Wang et al., 2024). Additionally, we explored enhancing the text using a large language model, *i.e.*, GPT-4 (Achiam et al., 2023), to generate longer descriptions without leveraging video information. The enhanced text and corresponding results are also presented in Figure 2 and Table 1. Our findings reveal that without the context provided by video, generating longer text does not effectively improve the retrieval performance, underscoring the critical role of video-specific information in enhancing text representations for retrieval tasks. However, our goal

---

[1]The vision encoder used here is the combination of BLIPv2's Q-former (Li et al., 2023a) and ViT (Dosovitskiy, 2020), the language decoder is LLAMA-2 (Touvron et al., 2023).

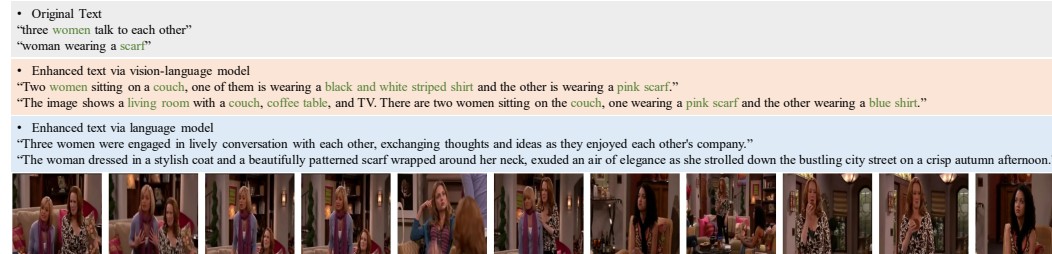

Figure 2: Comparison between original text, enhanced text generated by an image captioning model, and enhanced text produced by a language model. Original text and video are sourced from MSR-VTT dataset.

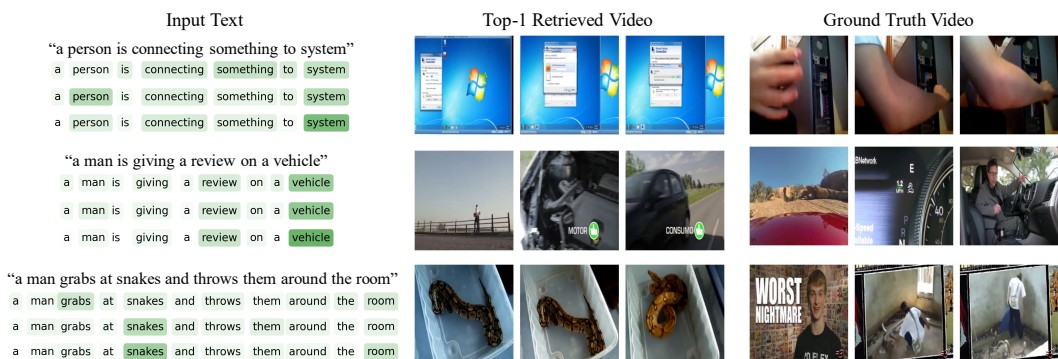

Figure 3: Visualization of attention scores across different layers in the text encoder of the ViT-B/32-based X-Pool model. We illustrate the attention maps from the 1st, 6th, and 12th layers, highlighting how attention distribution evolves throughout the text encoding process. Darker colors indicate higher attention scores.

is not to modify the text input itself, as in real-world applications, we cannot expect users to provide fully detailed descriptions. Therefore, in the next steps, we will explore how to leverage video information to enhance text representation, as this remains a crucial area of text-to-video retrieval.

### 3.3 FLAWED KEYWORD ATTENTION IN TEXT ENCODER

An input sentence consists of multiple words, and intuitively, focusing on different words can lead to varying interpretations of the sentence. Sometimes, even as humans, we need to carefully compare the text against several similar videos, repeatedly analyzing the text to find the correct text-video pair. In most retrieval frameworks, input texts are abstracted into latent features extracted by a text encoder. As discussed in Sec 3.1, the embedding $\mathbf{t}^{\langle eos \rangle} \in \mathbb{R}^d$ of the $\langle \text{EOS} \rangle$ token is commonly used as the representation of the entire text. Specifically, this representation is generated through the self-attention mechanism, which computes a weighted sum of all word embeddings in the sentence. We naturally hypothesize that a critical limitation of this approach lies in the self-attention mechanism of the text encoder, which, *if it inaccurately emphasizes certain words, can lead to a distortion of the text's intended meaning.* As illustrated in Figure 3, we visualize the attention scores of the text feature $\mathbf{t}^{\langle eos \rangle}$ across different layers in the text encoder. Specifically, we removed the scores of the $\langle \text{BOS} \rangle$ token and the $\langle \text{EOS} \rangle$ token itself, normalizing the remaining scores to ensure they sum to one. For instance, in the first example, the text encoder predominantly focuses on the word "system", resulting in a retrieved video that aligns more closely with this word than with the ground truth video, which should match the phrase "connecting something". In the third case, the encoder gives higher attention to the word "snakes," but words like "throw", "around", and "room" more accurately reflect the target video.

Additionally, we observe that the pretrained text encoder (Radford et al., 2021) tends to prioritize nouns, possibly due to biases from its pretraining dataset. However, in text-video retrieval, other words in the sentence, such as verbs and even prepositions, are often critical for identifying the correct video. Determining which words are truly keywords often requires guidance from the video content itself, beyond just the input sentence. This highlights the need to explore how video in-

formation can be leveraged to refine text representations, a direction that remains underexplored in previous works.

| MSR-VTT | | | | | | | | DiDeMo | | | | | | | |
|---|---|---|---|---|---|---|---|---|---|---|---|---|---|---|---|
| Avg | $k$=1 | $k$=2 | $k$=3 | $k$=4 | $k$=5 | $k$=6 | Attn | Avg | $k$=1 | $k$=2 | $k$=3 | $k$=4 | $k$=5 | $k$=6 | Attn |
| 30.9 | 31.2 | 32.1 | 32.9 | **33.5** | 32.4 | 32.2 | **33.0** | 24.8 | 24.6 | 24.9 | **25.2** | 25.1 | 25.0 | 24.6 | **25.6** |

Table 2: R@1 text-to-video retrieval results on MSR-VTT (Xu et al., 2016) and DiDeMo (Anne Hendricks et al., 2017) benchmarks, where 12 frames are uniformly sampled from each video. "Avg" means average pooling, "$k$" means top-$k$ pooling, and "Attn" represents the weighted sum approach.

### 3.4 IDENTIFYING KEY FRAMES FOR ADAPTING PRETRAINED IMAGE-TEXT MODELS

Given that the pretrained image-text model CLIP serves as the de facto encoder for extracting video representations $\{\mathbf{v}^1, \mathbf{v}^2, \ldots, \mathbf{v}^{\mathcal{T}}\}$, a persistent challenge in text-to-video retrieval is determining how to identify and fuse (Ni et al., 2022; Gorti et al., 2022; Luo et al., 2022; Bain et al., 2021) the most semantically relevant sub-regions of a video—represented as a subset of frames—that align with concise text descriptions. To illustrate the importance of key frames, we first conduct a toy experiment using the MSR-VTT and DiDeMo benchmarks, extracting text and frame-level features with the original pretrained CLIP encoders.

Given the text feature, we compare the results of three methods: (i) directly average pooling all the frame-level features; (ii) selecting the top-$k$ frames most similar to the text and averaging these $k$ features; and (iii) applying cosine similarity to assign weights for a weighted sum of all frames. Table 2 shows the corresponding results. From the results, we observe that: (i) a single frame is insufficient for satisfactory retrieval performance; (ii) treating all frames equally introduces irrelevant frames, leading to degraded performance; and (iii) identifying and leveraging key frames significantly improves results. As discussed in Sec. 3.3, we aim to utilize video information to refine the text representation. However, beyond the content-agnostic method of average pooling, the text feature must help identify key frames, using methods such as top-$k$ pooling or weighted sum.

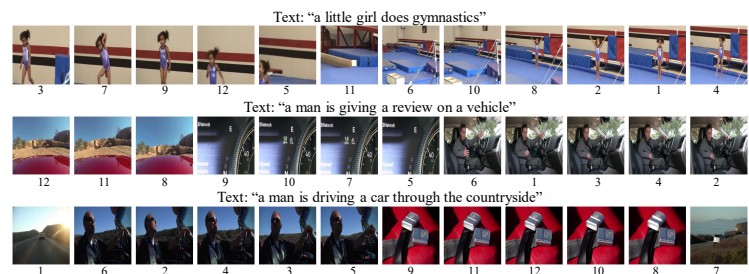

Figure 4: The similarity ranking between the text features and frame features, which are extracted by original pretrained CLIP encoder. We observe that the key frames, which are more relevant to the text description, make up a larger proportion of the video content.

Our hypothesis is that *key frames, or critical content, should comprise the majority of the video*. This is because text descriptions tend to focus on the most important information, which often occupies the largest portion of the video. To support this, we randomly selected several text-video pairs from the MSR-VTT dataset and ranked the frames by their similarity to the text feature (based on cosine similarity), as shown in Figure 4. Our findings indicate that the most similar "key frames" often constitute a significant portion of the video sequence, further validating our hypothesis. Accordingly, we propose a video abstraction method that relies solely on the video itself to fuse the frame-level features, without requiring additional modalities. This method will be introduced in detail in Sec. 4.2.

## 4 OUR WORK

In this section, we outline the key components of our proposed VICTER framework, designed for video-conditioned text representation refinement in text-to-video retrieval. We start with the basic feature extraction process in Sec.4.1, followed by the video abstraction method in Sec.4.2. Next, we detail the video-conditioned text enhancement mechanism in Sec.4.3. Finally, we present the overall architecture in Sec.4.4, as illustrated in Figure 7 in appendix.

## 4.1 FEATURE REPRESENTATION

The objective of text-to-video retrieval is to align text and video features within a shared latent space. We use CLIP (Radford et al., 2021) as the base model for extracting multi-modal representations (Gorti et al., 2022; Xue et al., 2022b). As introduced in Sec. 3.1, given a video consisting of hundreds of frames, the common approach is to sample $\mathcal{T}$ frames and input them into CLIP, generating $\mathcal{T}$ frame representations: $\{\mathbf{v}^1, \mathbf{v}^2, \ldots, \mathbf{v}^{\mathcal{T}}\}$. Let $\phi_v$ and $\phi_t$ represent CLIP's image and text encoders, respectively. The feature extraction process can be formulated as:

$$\mathbf{v}^i = \phi_v(\mathbf{Frame}\ i), i = 1, ..., \mathcal{T}; \quad \{\mathbf{t}^{\langle bos \rangle}, \mathbf{t}^1, ..., \mathbf{t}^{\mathcal{K}}, \mathbf{t}^{\langle eos \rangle}\} = \phi_t(\mathbf{Text}), \tag{3}$$

where $\mathbf{v}^i, \mathbf{t}^i \in \mathbb{R}^d$, and $\mathcal{K}$ denotes the number of words in the input text. Previous works typically use $\mathbf{t}^{\langle eos \rangle}$ as the representation of the entire text. However, as shown in Sec. 3.3, this approach often inaccurately emphasizes specific words, distorting the intended meaning of the text. This motivates us to revisit and utilize individual word representations in the following sections.

## 4.2 VIDEO ABSTRACTION MODULE

As discussed in Sec.3.2, we need a video feature to help enhance the text representation. Given $\mathcal{T}$ frame video features $\{\mathbf{v}^1, \mathbf{v}^2, \ldots, \mathbf{v}^{\mathcal{T}}\}$, it's intuitive to directly apply average pooling(Luo et al., 2022) to obtain an abstracted video representation $\mathbf{v}^{\mathrm{pool}}$. However, as discussed in Sec.3.4, this content-agnostic method cannot effectively capture the key information of the video, leading to sub-optimal results. In VICTER, we propose a self-content-aware video abstraction module, as shown in Figure7(b). Given $\mathbf{v} = [\mathbf{v}^1, \mathbf{v}^2, \ldots, \mathbf{v}^{\mathcal{T}}] \in \mathbb{R}^{\mathcal{T} \times d}$, we first compute the affinity score among all the video representations:

$$\bar{\mathbf{v}} = \frac{\mathbf{v}}{|\mathbf{v}|}, \quad \mathbf{A} = \bar{\mathbf{v}}\bar{\mathbf{v}}^{\top}, \quad \mathbf{S} = \sum_{i=1}^{\mathcal{T}} A_{:,i}, \tag{4}$$

where $\bar{\mathbf{v}}$ represents the normalized video feature vectors. The attention matrix $\mathbf{A} \in \mathbb{R}^{\mathcal{T} \times \mathcal{T}}$ is computed by the dot product of the normalized video features with their transpose. The final affinity score $\mathbf{S} \in \mathbb{R}^{\mathcal{T}}$ is then obtained by summing the attention matrix across the last dimension. Then, the abstracted representation is obtained as:

$$\mathbf{v}^{\mathrm{abs}} = \sum_{i=1}^{\mathcal{T}} s_i \cdot \mathbf{v}^i, \tag{5}$$

where $s_i$ denotes the affinity score corresponding to each frame $\mathbf{v}^i$, and $\mathbf{v}^{\mathrm{abs}} \in \mathbb{R}^d$ encapsulates the entire video through this content-conditioned process.

## 4.3 VIDEO-CONDITIONED TEXT ENHANCEMENT MODULE

After getting our abstracted video representation, we use it to enhance the text representation in our video-conditioned module, as shown in Figure 7(c). We first project the video embedding $\mathbf{v}^{\mathrm{abs}} \in \mathbb{R}^d$ into a single query $\mathbf{Q}_v \in \mathbb{R}^{1 \times d}$ and word embeddings $\mathbf{t} = [\mathbf{t}^1, \mathbf{t}^2, \ldots, \mathbf{t}^{\mathcal{K}}] \in \mathbb{R}^{\mathcal{K} \times d}$ into key $\mathbf{K}_t \in \mathbb{R}^{\mathcal{K} \times d}$ and value $\mathbf{V}_t \in \mathbb{R}^{\mathcal{K} \times d}$ matrices, where we set the size of the projection dimension the same as the model's latent dimension $d$. The projections are defined as:

$$\mathbf{Q}_v = \mathrm{LN}(\mathbf{v}^{\mathrm{abs}})\mathbf{W}_Q, \quad \mathbf{K}_t = \mathrm{LN}(\mathbf{t})\mathbf{W}_K, \quad \mathbf{V}_t = \mathrm{LN}(\mathbf{t})\mathbf{W}_V, \tag{6}$$

where LN denotes the layer normalization (Lei Ba et al., 2016), $\mathbf{W}_Q$, $\mathbf{W}_K$, and $\mathbf{W}_V$ are projection matrices in $\mathbb{R}^{d \times d}$. We employ dot product attention (Vaswani, 2017) to compute relevance weights between the abstracted video representation and each word in the text, which are used to aggregate the word embeddings:

$$\mathbf{z}_{\mathbf{t}|\mathbf{v}^{\mathrm{abs}}} = \mathrm{Attention}(\mathbf{Q}_v, \mathbf{K}_t, \mathbf{V}_t) = \mathrm{softmax}\left(\mathbf{Q}_v\mathbf{K}_t^{\top}/\sqrt{d}\right)\mathbf{V}_t, \tag{7}$$

where the query $\mathbf{Q}_v$ is derived from the abstracted video representation, guiding the attention to identify and weigh the most relevant words through the key $\mathbf{K}_t$. The value $\mathbf{V}_t$ encapsulates the word representations, enabling the selection and aggregation of key words based on the corresponding video context. To further enhance the aggregated word embeddings, we apply a linear transformation with a residual connection, incorporating the capacity of a feed-forward network:

$$\mathbf{z}_{\mathbf{t}|\mathbf{v}^{\mathrm{abs}}} = \mathbf{z}_{\mathbf{t}|\mathbf{v}^{\mathrm{abs}}} + \mathrm{Linear}(\mathrm{LN}(\mathbf{z}_{\mathbf{t}|\mathbf{v}^{\mathrm{abs}}})). \tag{8}$$

Finally, we combine this refined embedding with the original text representation, $\mathbf{t}^{\langle eos \rangle}$, from the text encoder to produce the final enhanced text feature, $\mathbf{t_e}$:

$$\mathbf{t_e} = \mathbf{t}^{\langle eos \rangle} + \lambda \cdot \mathbf{z_{t|v^{abs}}}. \tag{9}$$

where the $\lambda \in \mathbb{R}^d$ is a learnable hyperparameter. After obtaining the enhanced text representation $\mathbf{t_e}$, we apply a text-conditioned video aggregation method, X-Pool (Gorti et al., 2022), allowing $\mathbf{t_e}$ to attend to the most semantically relevant frames. This approach pools the final video representation, $\mathbf{v}^{\mathrm{xpool}}$, as illustrated in Figure 7(d) in appendix.

### 4.4 OVERALL ARCHITECTURE

Building on the foundation of adapting pre-trained VLMs, we present the detailed framework of VICTER built on CLIP and X-Pool models (Gorti et al., 2022) in Figure 7 in appendix. However, VICTER is versatile and can be seen as a general module for video-conditioned text representation refinement. For example, if we replace X-Pool's vision encoder with CLIP-VIP's vision encoder (Xue et al., 2022b), which directly uses the first video proxy token as the video representation, our video-conditioned text enhancement module can be easily integrated into their network without major adjustments. During the testing phase, the extra cost is from the cross-attention mechanism added during the text representation enhancement stage between each text-video pair, resulting in negligible additional computational overhead.

## 5 EXPERIMENT

### 5.1 EXPERIMENT SETUP

**Datasets.** We evaluate our method on three widely used text-video retrieval benchmarks:
- MSR-VTT (Xu et al., 2016) contains 10,000 video clips, each annotated with 20 sentences. We follow the 1K testing split used in (Gorti et al., 2022).
- DiDeMo (Anne Hendricks et al., 2017) comprises 10,642 video clips paired with a total of 40,543 captions. We adopt the train-test split from (Bain et al., 2021), where all sentence descriptions for a video are concatenated into a single query.
- LSMDC (Rohrbach et al., 2015) consists of 118,081 short clips from 202 movies, with each video typically paired with one caption. We follow the split defined by Gorti et al. (2022), using 109,673 videos for training, 7,408 for validation, and 1,000 for testing.

**Evaluation Metrics.** We assess model performance using standard retrieval metrics, including R@K (Recall at Rank K=1,5,10, higher is better), Median Rank (MdR, lower is better), and Mean Rank (MnR, lower is better), following the protocols from (Luo et al., 2022; Gorti et al., 2022).

**Implementation Details.** We initialize both the text and image encoders using CLIP checkpoints (ViT-B/32 and ViT-B/16). All experiments are conducted on a single NVIDIA A100 80GB GPU with PyTorch library. Following (Gorti et al., 2022), new FC layers are initialized with identity matrices, and biases are set to zero. Our models are fine-tuned end-to-end on each dataset, with 12 frames uniformly sampled from each video and resized to 224×224. We use a batch size of 32 for all experiments, setting the learning rate to 1e-6 for CLIP-initialized weights and 1e-5 for other parameters. The models are trained for 5 epochs (10 epochs for DiDeMo), optimized using the AdamW optimizer with a weight decay of 0.2 and a cosine learning rate schedule.

### 5.2 COMPARISON WITH STATE-OF-THE-ART METHODS

We show the text-to-video retrieval performance of our VICTER with previous methods across three benchmark datasets in Tables 3. The results reveal that VICTER significantly improves retrieval performance over traditional CLIP-based frameworks, such as X-Pool (Gorti et al., 2022), across all metrics. Specifically, VICTER enhances the ViT-B/32 based X-Pool at R@1 by 0.8% on the MSR-VTT and by 1.2% with the ViT-B/16 backbone. Additionally, when integrated with a post-pretraining framework that better utilizes temporal information in videos, VICTER further boosts the state-of-the-art results of the CLIP-VIP (Xue et al., 2022b) with the ViT-B/16 backbone on

| Method | MSR-VTT | | | | DiDeMo | | | | LSMDC | | | |
|---|---|---|---|---|---|---|---|---|---|---|---|---|
| | R@1↑ | R@5↑ | R@10↑ | MnR↓ | R@1↑ | R@5↑ | R@10↑ | MnR↓ | R@1↑ | R@5↑ | R@10↑ | MnR↓ |
| *CLIP-ViT-B/32* | | | | | | | | | | | | |
| CLIP4Clip (Luo et al., 2022) | 44.5 | 71.4 | 81.6 | 15.3 | 42.8 | 68.5 | 79.2 | 18.9 | 22.6 | 41.0 | 49.1 | 61.0 |
| X-CLIP (Ma et al., 2022) | 46.1 | 73.0 | 83.1 | 13.2 | 45.2 | 74.0 | - | 14.6 | 23.3 | 43.0 | - | 56.0 |
| TS2-Net (Liu et al., 2022) | 47.0 | 74.5 | 83.8 | 13.0 | 41.8 | 71.6 | 82.0 | 14.8 | 23.4 | 42.3 | 50.9 | 56.9 |
| X-Pool (Gorti et al., 2022) | 46.9 | 72.8 | 82.2 | 14.3 | 44.6 | 73.2 | 82.0 | 15.4 | 25.2 | 43.7 | 53.5 | 53.2 |
| X-Pool + VICTER (**Ours**) | 47.7 | 73.4 | 82.8 | 13.6 | 45.5 | 73.8 | 82.3 | 14.9 | 25.7 | 44.0 | 53.8 | 52.7 |
| UATVR (Fang et al., 2023) | 47.5 | 73.9 | 83.5 | 12.3 | 43.1 | 71.8 | 82.3 | 15.1 | - | - | - | - |
| Prompt Switch (Deng et al., 2023) | 47.8 | 73.9 | 82.2 | 14.1 | - | - | - | - | 23.1 | 41.7 | 50.5 | 56.8 |
| DiffusionRet (Jin et al., 2023b) | 49.0 | 75.2 | 82.7 | 12.1 | 46.7 | 74.7 | 82.7 | 14.3 | 24.4 | 43.1 | 54.3 | **40.7** |
| CLIP-ViP (Xue et al., 2022b) | 50.1 | 74.8 | 84.6 | 13.8 | 48.6 | 77.1 | 84.4 | 14.4 | 25.6 | 45.3 | 54.4 | 53.6 |
| CLIP-ViP + VICTER (**Ours**) | 50.5 | 75.1 | 84.8 | 13.4 | 49.0 | 77.1 | 84.6 | 14.0 | 26.0 | 45.5 | 54.2 | 53.0 |
| T-MASS Wang et al. (2024) | 50.2 | 75.3 | 85.1 | 11.9 | 50.9 | 77.2 | 85.3 | 12.1 | 28.9 | 48.2 | 57.6 | 43.3 |
| T-MASS + VICTER (**Ours**) | **50.8** | **75.7** | **85.3** | **11.6** | **51.4** | **77.5** | **85.4** | **11.8** | **29.9** | **48.8** | **57.9** | 42.8 |
| *CLIP-ViT-B/16* | | | | | | | | | | | | |
| X-CLIP (Ma et al., 2022) | 49.3 | 75.8 | 84.8 | 12.2 | 47.8 | 79.3 | - | 12.6 | 26.1 | 48.4 | - | 46.7 |
| TS2-Net (Liu et al., 2022) | 49.4 | 75.6 | 85.3 | 13.5 | - | - | - | - | - | - | - | - |
| X-Pool (Gorti et al., 2022) | 48.2 | 73.7 | 82.6 | 12.7 | 47.3 | 74.8 | 82.8 | 14.2 | 26.1 | 46.8 | 56.7 | 47.3 |
| X-Pool + VICTER (**Ours**) | 49.4 | 75.7 | 84.5 | 12.1 | 48.4 | 74.9 | 83.1 | 13.9 | 26.8 | 47.2 | 57.0 | 46.6 |
| UATVR (Fang et al., 2023) | 50.8 | 76.3 | 85.5 | 12.4 | 45.8 | 73.7 | 83.3 | 13.5 | - | - | - | - |
| CLIP-ViP (Xue et al., 2022b) | 54.2 | 77.2 | 84.8 | 11.3 | 50.5 | 78.4 | 87.1 | 12.8 | 29.4 | 50.6 | 59.0 | 43.1 |
| CLIP-ViP + VICTER (**Ours**) | **54.5** | **77.5** | 85.0 | 11.2 | 50.9 | 78.5 | 87.1 | 12.6 | 29.9 | 51.0 | 59.3 | 42.4 |
| T-MASS (Wang et al., 2024) | 52.7 | 77.1 | 85.6 | 10.5 | 53.3 | 80.1 | 87.7 | 9.8 | 30.3 | 52.2 | **61.3** | 40.1 |
| T-MASS + VICTER (**Ours**) | 53.6 | **77.5** | **85.8** | **10.1** | **53.9** | **80.5** | **87.7** | **9.6** | **30.7** | **52.4** | 61.2 | **40.0** |

Table 3: Text-to-video comparisons on MSRVTT (Xu et al., 2016), DiDeMo (Anne Hendricks et al., 2017) and LSMDC (Rohrbach et al., 2015). Bold number denotes the best performance.

| Method | MSR-VTT | | | DiDeMo | | |
|---|---|---|---|---|---|---|
| | R@1 | R@5 | MnR | R@1 | R@5 | MnR |
| Baseline | 46.9 | 72.8 | 14.3 | 44.6 | 73.2 | 15.4 |
| Average | 47.4 | 73.0 | 13.8 | 45.2 | 73.6 | 15.1 |
| Random | 47.0 | 72.7 | 14.0 | 44.5 | 73.5 | 15.3 |
| X-Pool | 46.5 | 71.9 | 14.9 | 44.0 | 72.7 | 15.4 |
| Abstract | **47.7** | **73.4** | **13.6** | **45.5** | **73.8** | **14.9** |

(a) Video abstraction methods.

| λ | DiDeMo | | |
|---|---|---|---|
| | R@1 | R@5 | MnR |
| Baseline ($\lambda=0$) | 44.6 | 73.2 | 15.4 |
| $\lambda=1$ | 43.8 | 72.5 | 16.0 |
| $\lambda=0.1$ | 45.4 | 73.5 | 14.9 |
| $\lambda=0.01$ | 45.2 | 73.3 | 15.3 |
| Learnable | **45.5** | **73.8** | **14.9** |

(b) Enhancement coefficient.

| Manner | DiDeMo | | |
|---|---|---|---|
| | R@1 | R@5 | MnR |
| Baseline | 44.6 | 73.2 | 15.4 |
| Concat | 45.3 | 73.3 | 15.0 |
| Multiply | 44.1 | 72.8 | 16.1 |
| Addition | **45.5** | **73.8** | **14.9** |

(c) Fusion manner.

Table 4: Ablation study on (a) different video abstraction methods; (b) the hyperparameter $\lambda$ in Eq. 9; and (c) different fusion manner.

MSR-VTT by 0.3%. This demonstrates that our video-conditioned text representation enhancement provides complementary improvements to existing CLIP-based text encoders, even when CLIP-VIP is further trained on additional datasets such as WebVid-2.5M and HD-VILA-100M. Moreover, when combined with T-MASS (Wang et al., 2024), which incorporates more variability in text embeddings, our VICTER model still achieves significant performance gains. This underscores the effectiveness of our proposed method in enhancing text-to-video retrieval performance.

In addition, we compare the video-to-text retrieval results with other methods in Table 5 in the appendix. Across various frameworks, our VICTER consistently enhances performance, achieving significant improvements in video-to-text retrieval.

## 5.3 ABLATION STUDIES

**Impact of video abstraction module.** To demonstrate the effectiveness of our proposed video abstraction module, we compare it with several variants, including mean pooling, random selection, and X-Pool-based selection. As shown in Table 4a, the X-Pool-based selection strategy, which uses the original text feature extracted by the encoder to perform weighted fusion of frame features, performs the worst. This is likely because the original text feature already contains inherent biases, and leveraging this biased feature to extract video information only amplifies these biases, leading to poorer performance. In contrast, average pooling outperforms random selection, indicating that most frames can accurately represent the video's content. Our abstraction method achieves the best results, supporting our hypothesis that emphasizing the content that occupies more of the video is a more effective strategy.

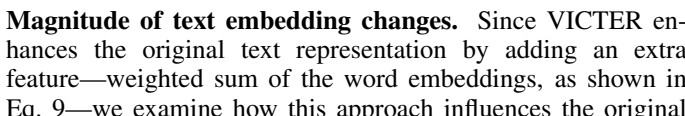

Figure 5: Visualization of attention scores in X-Pool's final text encoder (top row) and VICTER's text enhancement module (bottom row), alongside the corresponding video frames.

**Effect of learnable parameter.** Here, we ablate the hyperparameter $\lambda$ used in Eq.9, as shown in Table4b. The baseline ($\lambda$=0) represents the model without the text enhancement module. Our results demonstrate that an appropriate choice of $\lambda$ leads to improved performance, indicating the importance of carefully balancing the contribution of the extra representation. A learnable per-channel $\lambda$ achieves the best results, as it allows for fine-grained control over the size and influence of the added representation.

**Ablation on enhancement manner.** As shown in Table 4c, we explore three different fusion methods: concatenating the original and newly aggregated word embeddings followed by a linear layer to restore the channel dimension, multiplying the newly aggregated word embeddings with the original ones, and a simple addition. Our results indicate that the simple addition approach achieves the best performance.

**Weight of the key word.** We visualize the attention weights of each word in the input sentence using our video-conditioned text enhancement module, based on paired videos from the MSR-VTT test set, and compare these results to X-Pool's text encoder's final attention layer in Figure 5. It is evident that incorporating video information enables our model to capture more contextually rich words that better convey content and spatial relations, such as "hairdresser and client" or "bird in box," rather than solely focusing on nouns. However, there are still potential limitations. For instance, in the last example, "woman is cooking food" takes up more time in the video, causing our video abstraction module to prioritize these frames and extract keywords related to this part, while neglecting the later action "man is setting a table". This also highlights how our approach can complement original text encoder, balancing out each other's weaknesses.

**Magnitude of text embedding changes.** Since VICTER enhances the original text representation by adding an extra feature—weighted sum of the word embeddings, as shown in Eq. 9—we examine how this approach influences the original

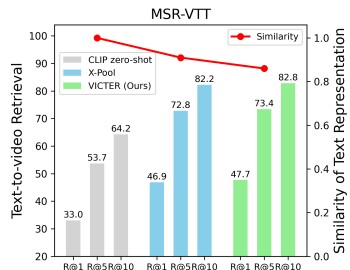

Figure 6: The comparison of the magnitude of text embedding changes between the original CLIP, X-Pool, and our VICTER is illustrated. We measure the differences in features using cosine similarity to indicate the extent of change between embeddings.

feature. In Figure 6, we display the cosine similarity between the VICTER-enhanced final text representation and the pretrained CLIP text representation on MSR-VTT test set. After fine-tuning with methods like X-Pool, the output from the text encoder still maintains a high similarity (0.91) compared to the zero-shot CLIP. However, with VICTER, the similarity between the text features and the original ones decreases, indicating that our model effectively captures new word-level information, which helps improve the overall retrieval performance.

## 6    CONCLUSION

This work focuses on refining text representation by addressing the inherent limitations of adapting pretrained image-text models for text-video retrieval tasks. We observed that directly using a pretrained text encoder often results in suboptimal bias, where the model tends to overemphasize certain nouns, leading to misinterpretations of the description. Moreover, key elements in the video, which typically occupy a significant portion of the frames, can serve as useful prior knowledge for extracting video feature. Based on these insights, we propose a versatile solution with two components: the video abstraction module and the video-conditioned text enhancement module. These modules can be seamlessly integrated into existing retrieval frameworks to significantly improve performance with minimal additional computational cost.

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

# A  APPENDIX

In this appendix, we show the detailed overall architecture of the "X-Pool + VICTER" method in Table 3, as well as the video-to-text retrieval results on MSR-VTT (Xu et al., 2016) dataset.

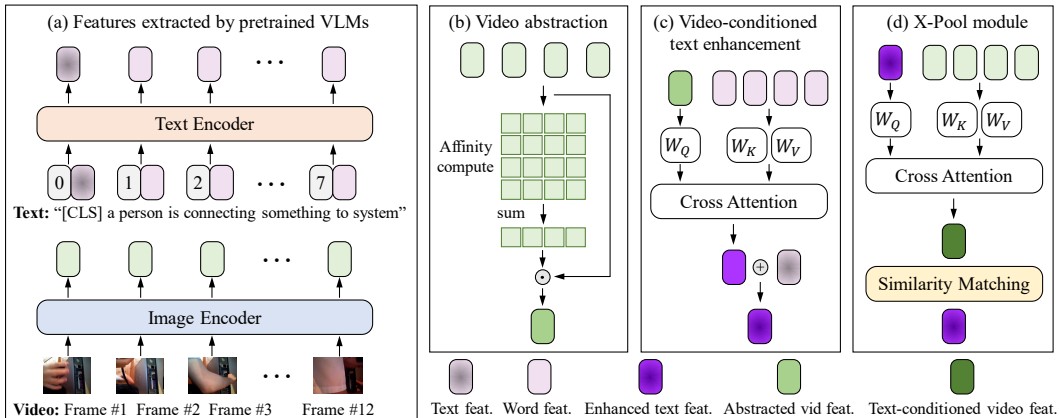

Figure 7: The detailed architecture of the proposed X-Pool + VICTER, *i.e.*, **vi**deo-**c**onditioned **te**xt representation **r**efinement, comprises several components: the basic pretrained VLMs (Sec.4.1), the video abstraction module (Sec.4.2), the video-conditioned text enhancement module (Sec. 4.3), and the text-conditioned pooling as introduced in (Gorti et al., 2022).

| Method | R@1↑ | R@5↑ | R@10↑ | MnR↓ |
|---|---|---|---|---|
| *CLIP-ViT-B/32* | | | | |
| CLIP4Clip (Luo et al., 2022) | 42.7 | 70.9 | 80.6 | 11.6 |
| X-Pool (Gorti et al., 2022) | 44.4 | 73.3 | 84.0 | 9.0 |
| X-Pool + VICTER (**Ours**) | 45.4 | 73.7 | 84.3 | 8.8 |
| TS2-Net (Liu et al., 2022) | 45.3 | 74.1 | 83.7 | 9.2 |
| DiffusionRet (Jin et al., 2023b) | 47.7 | 73.8 | 84.5 | 8.8 |
| UATVR (Fang et al., 2023) | 46.9 | 73.8 | 83.8 | 8.6 |
| T-MASS (Wang et al., 2024) | 47.7 | 78.0 | 86.3 | 8.0 |
| T-MASS + VICTER (**Ours**) | 48.5 | 78.4 | 86.5 | 7.9 |
| CLIP-ViP (Xue et al., 2022b) | 49.0 | 76.8 | 84.3 | 9.3 |
| CLIP-ViP + VICTER (**Ours**) | 49.3 | 77.1 | 84.7 | 9.0 |
| *CLIP-ViT-B/16* | | | | |
| X-Pool (Gorti et al., 2022) | 46.4 | 73.9 | 84.1 | 8.4 |
| X-Pool + VICTER (**Ours**) | 47.5 | 74.3 | 84.2 | 8.0 |
| TS2-Net (Liu et al., 2022) | 46.6 | 75.9 | 84.9 | 8.9 |
| UATVR (Fang et al., 2023) | 48.1 | 76.3 | 85.4 | 8.0 |
| T-MASS (Wang et al., 2024) | 50.9 | 80.2 | 88.0 | 7.4 |
| T-MASS + VICTER (**Ours**) | 51.5 | 80.4 | 87.9 | 7.2 |

Table 5: Video-to-text results on MSR-VTT dataset. (Xu et al., 2016).

