# OpenReview forum: "The Devil is in the Word: Video-Conditioned Text Representation Refinement for Text-to-Video Retrieval"
_ICLR.cc/2025/Conference — ICLR 2025 Conference Withdrawn Submission_

### Official Review · Reviewer_BUCE · 2024-10-31

**Soundness:** 3
**Presentation:** 2
**Contribution:** 2
**Rating:** 5
**Confidence:** 4

**Summary:**

The paper presents VICTER, a novel approach to enhance text representations for text-video retrieval tasks. While existing models like CLIP perform effectively, they often overlook critical details when relying solely on text to represent videos. VICTER addresses this by integrating video features in two stages: first, extracting key visual elements from video frames, followed by using these features to refine the text representation.

**Strengths:**

1. The motivation for this paper is well detailed in Section 3.
2. The proposed approach can be seamlessly integrated into existing frameworks, yielding consistent improvements.
3. The ablation study demonstrates the effectiveness of proposed modules.

**Weaknesses:**

1.  The paper presents some readability challenges. Section 4 includes excessive references to Section 3 and Figure 7, which is located in the Appendix.
2.  The overall contribution seems incremental, with the main observations and motivations similar to T-MASS [1].

Ref [1] Wang, Jiamian, et al. "Text Is MASS: Modeling as Stochastic Embedding for Text-Video Retrieval.” CVPR 2024.

**Questions:**

In section 3.2, the author shows that replace original text in the dataset with more informative descriptions can significantly boost retrieval performance. And the author claims that "in real-world applications, we cannot expect users to provide fully detailed descriptions". I am curious about the results of training with generated descriptions while testing with the original text.

---

### Official Review · Reviewer_qRpj · 2024-11-01

**Soundness:** 3
**Presentation:** 3
**Contribution:** 1
**Rating:** 3
**Confidence:** 5

**Summary:**

VICTER highlights the necessity of incorporating video content to generate robust text features. Preliminary experiments demonstrate the critical role of text features. The authors further analyze how biases within the text features negatively impact performance. To address this problem, they propose a video-conditioned transformer suitable for multiple existing methods.

**Strengths:**

This paper argues that text representations exhibit inherent biases, underscoring the need for video-conditioned methods. The idea is clear.

Detailed preliminary experiments effectively demonstrate the existing challenges.

The proposed modules can be integrated with various approaches.

**Weaknesses:**

The proposed module shows limited innovation, appearing primarily as a variant of Xpool. Specifically, Xpool is a text-conditioned transformer, while VICTER is adapted as a video-conditioned transformer. As shown in Figure 7, the overall framework appears to employ a bidirectional XPool.

There is a lack of comparison between video-conditioned transformers and text-conditioned transformers. Experimental evaluations of CLIP4Clip+video-conditioned transformer and CLIP4Clip+text-conditioned transformer are needed. Which is more effective?

ActivityNet is also a widely used benchmark, and experiments on it are essential for comprehensive evaluation.

The ablation study in Table 4bc lacks results on MSRVTT.

**Questions:**

Please see the weaknesses above.

---

### Official Review · Reviewer_tEeT · 2024-11-01

**Soundness:** 3
**Presentation:** 2
**Contribution:** 3
**Rating:** 6
**Confidence:** 4

**Summary:**

The paper tackles the task of text-video retrieval. More exactly, it argues that the textual representations alone are not expressive enough and may suffer from bias. In order to address this, the paper introduces VICTER, a method that enhances the textual representations using visual information. Finally, the authors test the proposed approach on multiple datasets.

**Strengths:**

The paper tackles an important problem and highlights some interesting aspects about the information contained in the textual features. It proposes an interesting method to address some of the problems and seems to achieve promising results.

**Weaknesses:**

The overall architecture section is very general and not easy to follow. In order to make it more clear, the architecture picture needs to be moved into the main part of the paper. While I appreciate the detailed analysis, one of the goals of the paper is to introduce VICTER and I think in the current form it's quite hard to follow what happens there. More exactly, section 4.4 needs more details on how all components interact one with the other. Please see more questions below for clarifications.

The limitations especially the computational cost I think needs more information. While I understand that the inference cost can be negligible since various parts can be pre-processed, I think it would make sense to have a proper discussion about the computational resources both at inference, but also during pre-processing.

Missing citations:
1. Mitigating Hubness in Cross-Modal Retrieval with Query and Gallery Banks](https://aclanthology.org/2023.emnlp-main.652) (Wang et al., EMNLP 2023)
2. Bogolin, Simion-Vlad, et al. "Cross modal retrieval with querybank normalisation." Proceedings of the IEEE/CVF Conference on Computer Vision and Pattern Recognition. 2022.

**Questions:**

What I don't understand is what video is used to enhance the textual query. Can you please elaborate? When computing the similarity between video_i and query_j, you enhance query_j with video_i and only with video_i?

---

### Official Review · Reviewer_4R5H · 2024-11-02

**Soundness:** 1
**Presentation:** 3
**Contribution:** 2
**Rating:** 3
**Confidence:** 5

**Summary:**

This paper proposes a video-conditioned text representation refinement method called VICTER for text-to-video retrieval. The authors aim to enhance the textual representation by using video information to highlight relevant words, thus improving retrieval accuracy. Specifically, the approach involves two key components: a video abstraction module that summarizes representative features from video frames and a video-conditioned text enhancement module that refines text embeddings based on these video features. The method shows state-of-the-art performance improvements on several benchmark datasets (MSR-VTT, DiDeMo, LSMDC).

**Strengths:**

1. The paper addresses an important aspect of text-to-video retrieval, focusing on enhancing text representation using video information, which is often overlooked compared to video feature improvement.

2. Empirical results demonstrate the effectiveness of the proposed method, achieving notable improvements over baseline models across multiple benchmark datasets.

3. The VICTER module can be integrated with various existing frameworks, showing its versatility in different retrieval tasks, including both image-text and video-text models.

**Weaknesses:**

1. The proposed video-conditioned text representation method appears impractical for real-world text-video retrieval systems. In actual inference scenarios, it is unrealistic to assume prior knowledge of the video-text pairing, making it computationally infeasible to conduct video-conditioned text generation for each candidate video. This suggests either a data leakage risk (if not independently enhancing text for each candidate) or an unacceptably high computational burden (if done independently for each candidate).

2. The video-conditioned text enhancement module, if applied independently for each candidate video, incurs significant computational overhead. This renders the approach infeasible for real-time or large-scale retrieval tasks where thousands of videos may need to be evaluated against a single text query.

3.  The paper lacks a clear explanation of how to avoid data leakage during inference. The text enhancement process involves generating text conditioned on the video, which could lead to unfair retrieval results if the same process is used during inference without independently processing each candidate video. This issue raises concerns about the validity of the reported improvements.

**Questions:**

1. How does the method handle the computational burden during inference when video-conditioned text enhancement is applied to potentially thousands of candidate videos? Is there any strategy to reduce the complexity in such cases?

2.  How does the model avoid data leakage during inference, given that the proposed method conditions text representation on the video? Are there measures in place to ensure that the video-specific text enhancement does not unfairly bias the retrieval process, namely, during the inference code, the ground truth is provided first for metric measure purpose?

---

### Official Review · Reviewer_36EG · 2024-11-04

**Soundness:** 2
**Presentation:** 2
**Contribution:** 2
**Rating:** 5
**Confidence:** 2

**Summary:**

The paper introduces VICTER (Video-Conditioned Text Representation Refinement), a method for text-to-video retrieval that addresses limitations in current vision-language models (VLMs) like CLIP. VICTER aims to improve text representations by integrating video context, using a video abstraction module and a video-conditioned text enhancement module. This approach helps refine text features by focusing on key video elements, leading to improved retrieval accuracy on benchmark datasets like MSRVTT, DiDeMo, and LSMDC.

**Strengths:**

* VICTER addresses a the limitation in pre-trained vision-language models by refining text features with video context, improving accuracy
* The paper conducts thorough preliminary studies that highlight the need for this approach, providing strong support for the proposed methodology.

**Weaknesses:**

* Clarity:
    * Figure 1 lack detailed explanations, and certain elements which may hinder reader comprehension of the technical processes.
    * The overview architecture is placed in the appendix rather than the main content, which is unconventional and could make it harder for readers to follow the core methodology.
* The model’s approach to generating and storing video abstractions may face storage bottlenecks as the database size scales up, potentially impacting its efficiency in large-scale retrieval tasks.

**Questions:**

The inference process for integrating video features into text is somewhat unclear. Could the authors clarify how the video-conditioned text refinement operates in practical use cases, especially in large-scale retrieval where individual video abstractions need to be integrated with text queries on the fly?

---

### Note · Authors · 2024-11-15

I have read and agree with the venue's withdrawal policy on behalf of myself and my co-authors.